# OpenReview forum: "SEAL: Safety-enhanced Aligned LLM Fine-tuning via Bilevel Data Selection"
_ICLR.cc/2025/Conference — ICLR 2025 Poster_

### Official Review · Reviewer_FGMp · 2024-10-29

**Soundness:** 4
**Presentation:** 4
**Contribution:** 3
**Rating:** 6
**Confidence:** 3

**Summary:**

This paper introduces SEAL, a novel framework to enhance safety for fine-tuning of large language models. Specifically, SEAL learns a sample ranker based on a bilevel optimization to up rank the safe and high-quality fine-tuning data and down rank the unsafe or low-quality ones. Extensive experiments on natural language processing benchmarks demonstrate the effectiveness of SEAL.

**Strengths:**

The research direction of this paper is crucial for safe model fine-tuning. The idea of the proposed SEAL seems to be simple and effective in addressing the issue of data ranking. The paper is well-written and easy to follow.

**Weaknesses:**

1. The reviewer noted a performance discrepancy between PEFT training for 7B models and full fine-tuning of 2.8B models. This might be due to the fact that only the k and v matrices are trained during PEFT training, which, while enhancing robustness to catastrophic forgetting, can amplify or diminish the impact of fine-tuning on safety. Therefore, It is  recommended that the authors conduct an ablation study comparing LoRA fine-tuning that incorporates weights in both the Feed Forward Network (FFN) and Multi-Head Self-Attention (MHSA) blocks with their current PEFT approach. Additionally, an analysis discussing how various LoRA configurations affect the safety-performance trade-off would be beneficial.

2. The author used the memory-efficient variant of SEAL by default in the experiments. To provide a more comprehensive understanding of the impact of this choice, I suggest the authors perform a quantitative comparison of memory consumption and performance between the standard and memory-efficient SEAL variants. Including a table or figure that illustrates this comparison would help clarify the effectiveness of both approaches.

**Questions:**

Please see the weakness section.

---

> ### Author Response · Authors · 2024-11-24
>
> Thank you for the careful review and support. Our response to your questions follows.
>
> **1. It is recommended that the authors conduct an ablation study comparing LoRA fine-tuning that incorporates weights in both the Feed Forward Network (FFN) and Multi-Head Self-Attention (MHSA) blocks with their current PEFT approach. Additionally, an analysis discussing how various LoRA configurations affect the safety-performance trade-off would be beneficial.**
>
> Thank you for the valuable suggestion. Indeed, the fine-tuning method can play a role in the performance discrepancy between different baselines and models. Due to the limitation of computational resource, in the main paper, we focused our experiments on the effectiveness and computational complexity of SEAL against other baselines. While an ablation study on how LoRA affects the performance of SEAL is certainly interesting and worth considering in the revision.
>
> **2. The author used the memory-efficient variant of SEAL by default in the experiments. To provide a more comprehensive understanding of the impact of this choice, I suggest the authors perform a quantitative comparison of memory consumption and performance between the standard and memory-efficient SEAL variants.**
>
> Great suggestion! Comparing the performance of the standard and memory efficient algorithm is the most important future direction of this branch of work. As compared to the memory efficient variant, the standard algorithm needs to additionally compute the value function $v(x)$ and its gradient. Currently, we find the standard algorithm difficult to run on our setup when the model size increases beyond 7 billion. This motivated us to propose the memory efficient variant which is more scalable to model size. In order to test standard algorithm, we believe additional implementation side techniques can be used. For example, one may offload the model to CPU when computing the value function $v(x)$ and its gradient. However, offloading can also increase the overall time complexity. It is indeed an interesting future direction, and can potentially further improves the performance of SEAL.

---

> > ### Author Response · Authors · 2024-12-02
> >
> > As a follow-up on our rebuttal, we would like to kindly remind the reviewer that the discussion period will conclude soon. We sincerely hope our response helps addressing the concerns, and we would greatly appreciate feedback from the reviewer.

---

### Official Review · Reviewer_TNmc · 2024-11-01

**Soundness:** 3
**Presentation:** 3
**Contribution:** 3
**Rating:** 6
**Confidence:** 4

**Summary:**

This work aims to enhance the fine-tuning LLMs’ safety capabilities while maintaining downstream performance. Therefore, authors design SEAN, a novel believes optimization framework to up-rank safe and high-quality fine-tuning data while down-ranking unsafe or low-quality data.  LLMs trained with SEAL demonstrate superior quality over multiple baselines, achieving an 8.5% and 9.7% increase in win rates compared to random selection on the LLAMA-3-8B-INSTRUCT and MERLINITE-7B models, respectively.

**Strengths:**

1. The paper is well-organized, and the methods and experiments are easy to understand.
2. The work attempts to address the safety concerns arising from LLM fine-tuning strategies, which is a very interesting and valuable issue.
3. The method of bilevel optimization is well-suited to the proposed problem.
4. The experiments roughly demonstrate the validity of the proposed method.

**Weaknesses:**

1. It would be better to provide more results, such as ablation studies. For example, comparing the method of fine-tuning the selector alone would be useful.
2. Some experimental results are not sufficiently explained.

**Questions:**

1. Could you compare the results of training the selector and LLMs separately? It is intuitive that this method seems to be more effective.
2. Could you further explain the results in Figure 3 and Figure 4. I am confused about why the performance of SEAL on the SLIMORCA dataset is not always superior to the baseline DSIR (Sometimes even worse).
3. From the current results, it is difficult to know whether the win rate improvements of SEAL are due to enhanced safety or better performance on downstream tasks. Could you provide more explanation about which datasets focus more on safety evaluation versus performance evaluation? Additionally, can you present more distinguishable results to show the model comparisons in terms of safety and performance, respectively?

---

> ### Author Response · Authors · 2024-11-24
>
> Thank you for the careful review and support. Our response to your questions follows.
>
> **1. Could you compare the results of training the selector and LLMs separately? It is intuitive that this method seems to be more effective.**
>
> We are sorry for the confusion. LLM fine-tuning and selector fine-tuning are already separate steps in SEAL workflow as described in Section 3.3. Specifically, we first train a data selector via solving the bilevel data selection formulation. Then note that we do not use the output model of the bilevel algorithm, we fine tune the aligned model on the task data filtered by the data selector.
>
> **2. Could you further explain the results in Figure 3 and Figure 4. I am confused about why the performance of SEAL on the SLIMORCA dataset is not always superior to the baseline DSIR (Sometimes even worse).**
>
> In Figure 3, SEAL outperforms DSIR on SlimOrca test dataset, while in Figure 4, SEAL is slightly outperformed by DSIR on SlimOrca test. The major reason is that performance of SEAL is dependent on the aligned model's architecture and parameters. Specifically, the aligned model determines the initial point of both SEAL's data selection training and the fine-tuning process. Since Figure 4 and 3 are tested on different models, variation in performance is expected. However, it is worth noting that SEAL is able to outperform DSIR in terms of average win rate on the three test datasets across all the models tested. We believe the average win rate on multiple datasets is a better metric than win rate on a single test dataset, as it showcases a model's capability on multiple domains more comprehensively.
>
> **3. From the current results, it is difficult to know whether the win rate improvements of SEAL are due to enhanced safety or better performance on downstream tasks.  Could you provide more explanation about which datasets focus more on safety evaluation versus performance evaluation? Additionally, can you present more distinguishable results to show the model comparisons in terms of safety and performance, respectively?**
>
> We are sorry for the confusion. In this work, we mainly use three test datasets: Anthropic Helpful and Harmless (HH) dataset, which contains a mixture of samples that can be used to evaluate either the helpfulness (instruction-following performance) or the harmlessness (safety) of the model; SlimOrca dataset, which contains instruction prompts and can be used to evaluate the task performance of the model; and HEx-PHI dataset containing red-teaming prompts that can be used to evaluate the safety of the model. Therefore, the model's fine-tuning performance and safety can be directly observed from the figures and plots in the main paper. For example, in Figure 3, we observe that SEAL is able to outperform all baselines on HH and HEx-PHI, and achieves similar performance to DSIR on SlimOrca. While in terms of average win rate on all test datasets, SEAL outperforms all baselines. Thus in this case we conclude that SEAL achieves comparable task performance to other methods while being able to better preserve safety. Overall, SEAL is able to output a stronger model.
>
> Thank you again for the careful review.

---

> > ### Comment · Reviewer_TNmc · 2024-11-26
> >
> > Thanks for the feedback. The experimental process has been clarified. However, the adequacy of the experiments still seems insufficient, so I will keep the original score unchanged.

---

### Official Review · Reviewer_fDtq · 2024-11-04

**Soundness:** 3
**Presentation:** 3
**Contribution:** 3
**Rating:** 5
**Confidence:** 3

**Summary:**

This paper proposed a security-enhanced fine-tuning framework for large language models, called SEAL, which aims to address the problem of negative impact on model security alignment during the fine-tuning process. By learning a data selector based on two-layer optimization, the fine-tuning data is ranked to enhance the weight of safe and high-quality data and reduce the weight of unsafe or low-quality data, thus enhancing the safety during the fine-tuning process. SEAL is tested on multiple models and datasets, and the results show its advantages in terms of validity, flexibility, and interpretability, and compared to baseline methods such as random selection, it has a improvement in the different models with a increase in win rate and maintains good performance with a range of data selection percentages, while the data selector is transferable and reduces computational complexity.

**Strengths:**

1. The paper proposes a security-enhanced fine-tuning framework named SEAL, specifically designed to mitigate the negative impacts on model security alignment during the fine-tuning process.
2. SEAL employs a novel two-layer optimization method to learn a data selector, which effectively ranks fine-tuning data. This process increases the emphasis on safe and high-quality data while reducing the influence of unsafe or low-quality data, thus improving fine-tuning safety.
3. The data selector used in SEAL is transferable across different contexts, reducing computational complexity and enhancing its applicability.

**Weaknesses:**

1. Lack of novelty. This paper presents a method for training data selectors and weighting losses, which is very straightforward. At the same time, the paper spends a lot of time on the optimization algorithm, but the optimization algorithm itself lacks improvement and innovation compared to the referenced papers. Overall, the article's approach lacks novelty, both in the idea itself and in the optimization method.
2. For all experiments, there was no adjustment for the variance of performance given after randomization of seeds. Especially for the randomized selection method, a large number of replicated experiments are necessary. Also, there are sampling methods for the generation of large language models, and randomness is introduced here as well. Overall, the lack of repeated experiments leads to unconvincing results.
3. Apart from the baseline chosen for this paper, I don't know why the authors didn't design additional but direct baselines, such as directly letting the model be given weights or rankings through the prompt engineering.
4. The results now lack insight, for example, the HEX-PHI data actually contains 11 categories, and it would be interesting to analyze the different categories in detail and explore the limitations of the sample selection algorithm, or an analysis on other benchmark tasks to explore the balance between security and performance would be insightful. A more in-depth analysis will help us better understand the effectiveness and flaws of the method, which is very meaningful.
5. LoRA introduces additional parameters, so it is difficult to directly attribute for performance improvements. Therefore, I believe that full parameter fine-tuning should be used for all methods. Also, the models do not perform well. Sometimes it is not even better than Random.

**Questions:**

See Weakness.

---

> ### Author Response · Authors · 2024-11-24
>
> Thank you for the careful review. Our response to your questions follows.
>
> **1. Overall, the article's approach lacks novelty, both in the idea itself and in the optimization method.**
>
> We respectfully disagree. The use scenario of SEAL is LLM fine-tuning, which is different from the vision tasks traditionally performed in previous bilevel works (e.g., [1][2]). Due to the different data format, loss types, model architectures and even evaluation process, it was unclear how the traditional bilevel data re-weighting approach can be used in LLM training, let alone effectively.  Due to the immense size of the language models, we find that the existing method in [1][2] are inefficient in LLM tasks. Therefore, we proposed more computationally efficient variants in Algorithm 1&2 where we utilized a fully single-loop structure and saved the computational cost of computing the value function's gradient. In the experiments, we showcase the effectiveness of bilevel LLM data selection through both win rate comparison and quality analysis for the first time. We also tested the transferability of the data selectors, which was not present in previous bilevel works.
>
> ```
> [1] Luca Franceschi, Michele Donini, Paolo Frasconi, and Massimiliano Pontil. Forward and reverse gradient-based hyperparameter optimization. arXiv:1703.01785, 2017.
>
> [2] Han Shen and Tianyi Chen. On penalty-based bilevel gradient descent method. ICML, 2023.
> ```
>
> **2. For all experiments, there was no adjustment for the variance of performance given after randomization of seeds. Especially for the randomized selection method, a large number of replicated experiments are necessary.**
>
> For the random selection baselines, the results reported in the paper are averaged over three random seeds. For other baselines, we are conducting experiments to test the variance between runs, and will share the results in future versions.
>
> **3. I don't know why the authors didn't design additional but direct baselines, such as directly letting the model be given weights or rankings through the prompt engineering.**
>
> Prompting LLMs to output absolute weights or rankings might have some variance and context length issues, while it is interesting to consider efficient implementations of such methods. We will consider adding them in the revisions.
>
> **4. The results now lack insight, for example, the HEX-PHI data actually contains 11 categories, and it would be interesting to analyze the different categories in detail and explore the limitations of the sample selection algorithm.**
>
> Thank you for the suggestion. In the revised paper, we have added Section (B.5) where the categorized win rate comparison on the Llama-3-8B-Instruct model is provided, Hope this resolves the question.
>
> **5. LoRA introduces additional parameters, so it is difficult to directly attribute for performance improvements. Therefore, I believe that full parameter fine-tuning should be used for all methods. Also, the models do not perform well. Sometimes it is not even better than Random.**
>
> We have tested full-parameter in Section B.1 on the Pythia-2.8b model. Only for the larger model like Llama-3, we use LoRA to save computational cost. It is worth noting that LoRA is widely used to fine-tune larger models like Llama family [1][2], and it has been observed in these works that the performance degradation is minimal as compared to full-parameter fine-tuning.
>
> Regarding the performance compared to random, it is worth noting that across all models tested, our method consistently outperforms random selection in terms of win rate averaged over three test datasets. Compared to random, SEAL’s average performance gain on LLAMA-3-8B-INSTRUCT is around 8.5% in Table1, and is around
>  9.8% on MERLINITE-7B in Table 2. On specific test datasets like HEx-PHI, SEAL outperforms random by 10\% on both Merlinite-7b and Llama-3-8b, and is only 1\% worse than random in either benign tests on Llama-2-7b or small model tests on Pythia-2.8b. While we argue that average test performance should be preferred over performance on one single test dataset, as it more comprehensively demonstrates a model's capability over multiple domains.
>
> ```
> [1] CY Hsu, YL Tsai, CH Lin, PY Chen, CM Yu, CY Huang. Safe LoRA: the Silver Lining of Reducing Safety Risks when Fine-tuning Large Language Models. NeurIPS, 2024.
> [2] Xiangyu Qi, Yi Zeng, Tinghao Xie, Pin-Yu Chen, Ruoxi Jia, Prateek Mittal, and Peter Henderson. Fine-tuning aligned language models compromises safety, even when users donot intend to! arXiv preprint arXiv:2310.03693,2023.
> ```
>
> Thank you again for the careful review.

---

> > ### Author Response · Authors · 2024-12-02
> >
> > As a follow-up on our rebuttal, we would like to kindly remind the reviewer that the discussion period will conclude soon. We hope our response helps addressing your concerns. If further clarification is needed, please do not hesitate to ask.

---

### Official Review · Reviewer_A6sy · 2024-11-05

**Soundness:** 3
**Presentation:** 2
**Contribution:** 3
**Rating:** 6
**Confidence:** 4

**Summary:**

This paper propose a data selection algorithm to protect the safety alignment of the LLM model after supervised finetuning. The proposed method is based on a bi-level optimization process, where the data are ranked by a learned ranker, such that the data points aligning with the safe dataset receives a higher rank. Experiment results show the proposed metho achieves both improved safety and task performance comparing to full supervised finetuning and random selection on various models.

**Strengths:**

This paper investigates an important topic: safety alignment in the supervised finetuning process of the LLM. The problem is a valid concern and the proposed data selection method is well-motivated. Although data selection is not novel by itself, the proposed method appears to be more straightforward, computation-efficient, and more effective comparing to previous methods. Experiments are conducted on various datasets and models, showing the transferability of the selected data across the models and the generalizability across the dataset. The proposed method appears to be overall effective and useful.

**Weaknesses:**

One potential drawback of this paper is it assume the safe dataset is readily available in the supervised finetuning process. With the safe dataset available, a straightforward idea would be directly optimize the model with a combination of safe data and task data, as formulated in Equation (4). Since this model update is also performed in the data selector training anyway, it is unclear if the data selection is realy needed to achieve improved safety. On the other hand, from algorithm 2 the data selector update can be based on the current model weight only. This leads to another important ablation study that we use the pretrained safe-aligned model for the loss compression directly, without further updating its weight in the data selection process. These two aspects will show if the proposed bi-level optimization is truly important to achieve the desired performance.

Besides the issue with safe data usage, the comparison against baseline data selection methods are also on the weak side. Multiple related work are listed in Sec. 2 that perform data selection for LLM safety. Yet in the experiments, only 1 data selection method, DSIR, and one additional dataset method are compared as baseline. More explaination is needed on the difference and benefit of the proposed method comparing to previous data selection methods mentioned in Sec. 2.

**Questions:**

1. Since a safe dataset is already availlable in the training, why not just train the model with the objective in Equ. (4) to balance safety and task objective, without further effort on doing data selection?

2. Why does the target domain win rate also improve as we select data based on safety alignment?

---

> ### Author Response · Authors · 2024-11-24
>
> Thank you for the careful review. Our response to your comments follows.
>
> **Weakness: One potential drawback of this paper is it assume the safe dataset is readily available in the supervised finetuning process. With the safe dataset available, a straightforward idea would be directly optimize the model with a combination of safe data and task data.**
>
> Our method is designed to filter the unsafe or low-quality samples in the task/fine-tuning data, and output a better task dataset. We prove this point by showing in the experiments that the model fine-tuned on SEAL-selected task dataset is better across several test datasets than the model fine-tuned on the original task dataset. Then combining the safe data with the SEAL-selected task dataset will improve over simply combining the safe dataset and the unfiltered task dataset.
>
> To support this point, within the limited time of the rebuttal period, we have conducted the suggested ablation study.
>
> |                                           | Anthropic HH | SlimOrca | HEx-PHI | Average    |
> | ----------------------------------------- | ------------ | -------- | ------- | --- |
> | Safety+fine-tuning dataset (0\% non-safe) | 50           | 50       | 50      |  50   |
> | Safety+SEAL                               | 51.25        | 49.69    | 50.63   |   50.52  |
>
> |                                            | Anthropic HH | SlimOrca | HEx-PHI | Average    |
> | ------------------------------------------ | ------------ | -------- | ------- | --- |
> | Safety+fine-tuning dataset (10\% non-safe) | 50           | 50       | 50      | 50    |
> | Safety+SEAL                                | 53.12        | 52.2     | 55.76   |  53.69   |
>
> |                                            | Anthropic HH | SlimOrca | HEx-PHI | Average    |
> | ------------------------------------------ | ------------ | -------- | ------- | --- |
> | Safety+fine-tuning dataset (50\% non-safe) | 50           | 50       | 50      |  50   |
> | Safety+SEAL                                | 62.5     | 55.93     | 58.5   |   58.98  |
>
> Due to the time limitation, we conducted these experiments on the Pythia-1b model. The experimental setup is the same as that in Section (B.1). In the table, **safety+fine-tuning dataset (x\% non-safe)** represents fine-tunng on the combination of safety dataset and the fine-tuning dataset which contains of x\% unsafe samples, and  **Safety+SEAL** is fine-tuning on the combination of safety dataset and SEAL-selected fine-tuning dataset.
> The evaluation metric is the win rate against **safety+fine-tuning dataset (x\% non-safe)** (see Section A.2 for description of win rate). The model is considered better if the win percent is higher. It can be observed that SEAL is able to consistently outperform directly combining the safety and task dataset without selection, across different unsafe ratios. This is because SEAL's data selection gets rid of the unsafe samples completely in the fine-tuning process, while directly combining safety data and task data will always suffer from the influence of these unsafe samples.
>
> **Q1. Since a safe dataset is already available in the training, why not just train the model with the objective in Equ. (4) to balance safety and task objective, without further effort on doing data selection?**
>
> This is because Equ. (4) still involves the unsafe samples, thus optimizing on a combination of safe data and non-selected task data can still lead to sub-optimal performance. In the paper, we show with win rate comparison in Section 4 and quality analysis in Section (B.2)&(B.3) that the SEAL is able to pick out unsafe samples in the task dataset. As a result, training with the filtered task data yields improved performance compared to training with the original task dataset, which is further supported by the added ablation study in our response to the previous question.
>
>
> **Q2. Why does the target domain win rate also improve as we select data based on safety alignment?**
>
> The reason is that seal is also able to filter the low-quality samples, e.g., see the quality analysis in Section (B.2) in the appendix, where SEAL filters the samples that have improper target answers. Since fine-tuning on these samples will damage the model's performance in target domain, filtering them out helps the performance.
>
>
> Thank you again for the careful review. We hope that our response addresses your questions, potentially leading to a higher score. Please let us know if you have further questions.

---

> > ### Comment · Reviewer_A6sy · 2024-11-25
> >
> > I would like to thank the author for the additional results. This resolves my concern on the necessity of the proposed data selection method. I will increase my score.

---

### Official Review · Reviewer_7ky2 · 2024-11-05

**Soundness:** 2
**Presentation:** 2
**Contribution:** 2
**Rating:** 6
**Confidence:** 4

**Summary:**

- Introduces SEAL, a novel bilevel fine-tuning framework that optimally balances fine-tuning data selection to enhance model safety, even when incorporating external non-safety-aligned data.

- Demonstrates SEAL’s robust performance improvements across diverse models and datasets, outperforming existing baselines in both safety and task-specific effectiveness.

- Provides a transferable and computationally efficient solution, with open-source code to support broader adoption and further research within the community.

**Strengths:**

I believe that the proposed fine-tuning methodology for enhancing LLM safety addresses a timely and critical issue in AI, making it a highly suitable topic for ICLR.

Moreover, this method has the potential to extend beyond the safety domain, offering a general and versatile approach applicable to various key areas, thereby holding significant value for the broader AI community.

The learning and evaluation code is transparently and accessibly provided, and with further refinement post-acceptance, it can greatly benefit researchers and practitioners through ICLR, fostering impactful contributions to the field.

**Weaknesses:**

(1)

Comment: One of the most significant weaknesses of this study lies in the lack of a fundamental explanation for why SEAL achieves performance improvements beyond using safety-aligned data alone. For instance, if the fine-tuning dataset consists entirely of non-safety-aligned data, it remains unclear why SEAL, when using 𝛾>0, would still mitigate performance degradation in safety compared to a setup with 𝛾=0. Ideally, both should yield comparable safety performance. To address this, the authors should provide comprehensive and detailed reasoning, supported by thorough analyses. Specifically, experiments varying the proportion of non-safety-aligned data from 0% to 100% should be conducted to illustrate how SEAL performs across different scenarios. This would demonstrate the method's robustness and technical validity, thereby satisfying readers and reviewers with a more rigorous and convincing evaluation.

Suggestion to authors:

- An ablation study varying the proportion of safety-aligned vs non-safety-aligned data in the fine-tuning set to demonstrate SEAL's effectiveness across different scenarios (e.g.,  varying the proportion of non-safety-aligned data from 0% to 100%).
- A theoretical analysis or intuitive explanation for why SEAL is effective even when the fine-tuning dataset contains non-safety-aligned data.
- Experiments comparing SEAL (𝛾>0) to a baseline (𝛾=0) across different ratios of safety-aligned to non-safety-aligned data.

(2)

Comment: The paper lacks a thorough investigation of similar approaches, such as bilevel optimization techniques applied to LLMs, and does not sufficiently highlight its relative novelty. Methods like self-supervised learning or unlearning can be viewed as forms of bilevel or similar hierarchical optimization.

Suggestion to authors:

- Provide a more comprehensive literature review section comparing SEAL to recent bilevel optimization approaches in LLMs, specifically highlighting the key differences and novelties of SEAL.
- Discuss how SEAL relates to or differs from self-supervised learning and unlearning methods in the context of LLM safety enhancement.
- Include a detailed comparison table or section that clearly outlines the methodological contributions of SEAL relative to existing approaches.

===


If this major concern is adequately addressed and clearly communicated during the review process, I would be open to reconsidering my score.

**Questions:**

Please refer to my comment for Weakness section.

---

> ### Author Response · Authors · 2024-11-24
>
> Thank you for the valuable feedback. Our response to your comments follows.
>
>
> **1. An ablation study varying the proportion of safety-aligned vs non-safety-aligned data in the fine-tuning set to demonstrate SEAL's effectiveness across different scenarios (e.g., varying the proportion of non-safety-aligned data from 0% to 100%); Experiments comparing SEAL (𝛾>0) to a baseline (𝛾=0) across different ratios of safety-aligned to non-safety-aligned data.**
>
> Thank you for the great suggestions. Within the limited time of rebuttal period, we have conducted the suggested ablation studies and added the new Section (B.4) in the revised paper. For convenience, we also present the results and discussion below.
>
>
>
> |                                           | Anthropic HH | SlimOrca | HEx-PHI | Average    |
> | ----------------------------------------- | ------------ | -------- | ------- | --- |
> | Safety+fine-tuning data (0\% non-safe) | 50           | 50       | 50      |  50   |
> | Safety data only ($\gamma$=0)                      | 49.69        | 47.81    | 50.9    |  49.47   |
> | Safety+SEAL         ($\gamma$>0)                       | 51.25        | 49.69    | 50.63   |   50.52  |
>
> |                                            | Anthropic HH | SlimOrca | HEx-PHI | Average    |
> | ------------------------------------------ | ------------ | -------- | ------- | --- |
> | Safety+fine-tuning data (10\% non-safe) | 50           | 50       | 50      | 50    |
> | Safety data only    ($\gamma$=0)                      | 50.5        | 49.37    | 56.06   |  51.98   |
> | Safety+SEAL      ($\gamma$>0)                           | 53.12        | 52.2     | 55.76   |  53.69   |
>
> |                                            | Anthropic HH | SlimOrca | HEx-PHI | Average    |
> | ------------------------------------------ | ------------ | -------- | ------- | --- |
> | Safety+fine-tuning data (50\% non-safe) | 50           | 50       | 50      |  50   |
> | Safety data only   ($\gamma$=0)                       | 60.31        | 52.5    | 61.25   |  58.02   |
> | Safety+SEAL   ($\gamma$>0)                               | 62.5     | 55.93     | 58.5   |   58.98  |
>
> We run these experiments on Pythia-1b model with the same setup as that in Section (B.1). In the table, **safety+fine-tuning dataset (x\% non-safe)** represents fine-tuning on the upper-level safety dataset combined with the original fine-tuning dataset with x\% of non-safety-aligned samples; **Safety dataset only ($\gamma$=0)** represents fine-tuning on the upper-level safety dataset only, so it is equivalent to SEAL with $\gamma$=0; and **Safety+SEAL($\gamma$>0)** is fine-tuning on the mixture of upper-level safety dataset and SEAL-selected fine-tuning dataset. The metric used in this table is the win rate (see Section A.2 for description of win rate) against **safety+fine-tuning dataset (x\% non-safe)** under each non-safe ratio. The model is considered better if the win rate is higher.
>
> It can be observed that when the fine-tuning dataset is completely safe with $0\%$ unsafe ratio, only using the safe dataset ($\gamma$=0) has an disadvantage on the fine-tuning domain (SlimOrca) since the other two baselines additionally utilize the safe fine-tuning dataset. When the unsafe ratio is 10\%, SEAL is able to achieve a noticeable average win rate advantage over the baselines. Although $\gamma=0$ improves performance on safety domain (HEx-PHI), it has worse performance on the fine-tuning domain (SlimOrca) than SEAL with $\gamma>0$. This is because $\gamma=0$ fails to utilize the safe subset of fine-tuning dataset, while SEAL is able to select a safe fine-tuning subset relatively well, which can be observed by its performance gain over no-selection. Similar observation can be made when the unsafe ratio increases to 50\%, $\gamma=0$ is outperformed by SEAL ($\gamma>0$) on the fine-tuning's target domain (SlimOrca). It is worth noting that our goal is to improve the model's capability on the fine-tuning domain while preserving its safety as much as possible, we can see SEAL helps achieving this goal. Lastly, we would like to clarify that SEAL belongs to the data selection method family. The use scenario for data selection methods assumes at least part of the samples in the original fine-tuning dataset is usable. An example of such a use case is fine-tuning on large datasets that contain useful information but may not be entirely safe, and thus needs filtering. In this case, a 100\% unsafe ratio leads to no motivation of fine-tuning at all.  Therefore, 100\% unsafe ratio is out-of-scope generally for all data selection methods.

---

> > ### Author Response · Authors · 2024-11-24
> >
> > **2. The paper lacks a thorough investigation of similar approaches, such as bilevel optimization techniques applied to LLMs.**
> >
> > Thank you for the valuable suggestion. In the revised paper, we have added more related works on bilevel optimization for LLMs and a discussion in Section 2, where we emphasize the differences of this work in application scenario and algorithmic from the bilevel self-supervised learning and bilevel unlearning works.
> >
> > We appreciate the thorough review and sincerely hope that our response addresses your questions, potentially leading to a higher score. Please let us know if additional clarification is needed.

---

> > > ### Comment · Reviewer_7ky2 · 2024-11-25
> > >
> > > Thank you for the additional experiments and analyses provided by the authors. However, the "Related Work" section does not clearly articulate the differences, advantages, and limitations compared to prior studies, which constrains the perceived novelty and contribution of this work. As a result, I remain uncertain if the paper is strong enough for acceptance at ICLR.
> > >
> > > While many existing studies have proposed so various data selection methodologies (e.g., https://arxiv.org/abs/2002.08484 as a typical example) and others have explored similar concepts within the scope of LLM applications (e.g., https://dl.acm.org/doi/pdf/10.1145/3626772.3657807), this work does exhibit some novelty by applying data selection methods (via bilevel optimization) to LLM-based frameworks. However, the innovation seems limited to the application of LLMs rather than introducing significant advancements in core technical methodologies.
> > >
> > > To strengthen the work, it is crucial to provide a broader and more detailed comparative analysis that either goes beyond the scope of LLM applications or thoroughly contrasts this approach with existing methodologies within the LLM domain. The inclusion of complete comparison results and a more comprehensive literature review is essential. Substantial improvements to the "Related Work" section are necessary to better position the novelty and contributions of this paper.

---

> > > > ### Author Response · Authors · 2024-11-27
> > > >
> > > > Thank you for the valuable suggestion. We have updated the related works section with the suggested paper and more discussions that contrast our work to other data selection and bilevel methods. We highlighted two major differences of this work from other data selection methods: 1) as compared to gradient similarity based data selection (e.g., https://arxiv.org/abs/2002.08484) that do not necessarily result in sufficient decrease in target safety loss (see, e.g., https://arxiv.org/pdf/2401.12926) and thus can be suboptimal in terms of safety, this work solves the bilevel data selection formulation that explicitly aims to minimize safety loss as much as possible after data selection; 2) compared to other bilevel optimization works that study data selection, this work focuses on LLM safety and propose more computationally efficient algorithms. In an effort to further reduce overall time complexity, we also conducted transferability test for our bilevel method, which was not present in previous works.
> > > >
> > > > Hope our response helps resolving your concerns. Let us know if further clarification is needed.

---

> > > > > ### Comment · Reviewer_7ky2 · 2024-11-27
> > > > >
> > > > > I would like to commend the authors for their additional efforts. While I still find that the study lacks rigorous comparative analysis with related work and does not sufficiently highlight its novelty, its timely relevance and the simplicity and effectiveness of its methodology represent valuable contributions to the community especially with the reproducible public code. Based on these merits, I have adjusted my score as an exception.
> > > > >
> > > > > If accepted, I strongly recommend including all experimental results mentioned in the discussion in the appendix. Additionally, providing more detailed and clear open-source code would greatly support follow-up research within the community. Expanding comparisons with related works and reinforcing the study's novelty through diverse reasoning or comparative experiments would further enhance its impact. If possible, these improvements should be included in the appendix before the camera-ready submission.
> > > > >
> > > > > ----------------------------
> > > > >
> > > > > However, that said, I still have concerns about the novelty (e.g., safety and bi-level context may be fundamentally more limited compared to other data selection methods) and the lack of deep comparative analysis with related work. Therefore, I could agree with any final decision from the AC and PC. Thank you.

---

> > > > > > ### Author Response · Authors · 2024-11-28
> > > > > >
> > > > > > Thank you for raising the score and the suggestions. We are happy to conduct the experiments in the discussion and include them in the camera-ready reversion if provided. We have also uploaded our code to anonymous github (https://anonymous.4open.science/r/seal_anonymous-C8DD) and the public repo will be included in the final revision.

---

### Public Comment · ~Tiansheng_Huang1 · 2024-11-26
**My two cents**

Hi,

I am a researcher who is actively working on the harmful fine-tuning attack that this paper aims to solve. I have carefully read the paper and the review. From the review, It seems that the main concern lies in its novelty and contribution, and I would like to express my view point as follows.

* The core contribution of SEAL I think lies in the problem formulation part. The authors design a bi-level problem to optimize the data selector, such that with this data selector we can filter out the harmful instances within the fine-tuning data, and retrain the model. The problem formulation is elegant and can be understood immediately by looking at Equation 1.

* Indeed, for how to solve the bi-level problem, the penalty solution seems to be ordinary. However, I think the main contribution of this paper is not to propose fancy solutions for classical bi-level problem solving but lies in how to design a **usable and interpretable** solution for the arising harmful fine-tuning attack.  The problem formulation itself is the most valuable contribution.

Personally, this is one of the good papers that I would like to pay a visit to if it can be appeared at the conference. Therefore, I would like to express my appreciation here.

Thanks,

Tiansheng Huang

https://huangtiansheng.github.io/

---

### Meta-Review · Area_Chair_tZkj · 2024-12-23

**Metareview:**

This work proposes a novel framework (SEAL) to enhance safety in LLM fine-tuning. SEAL learns a data ranker based on the bilevel optimization to up rank the safe and high-quality fine-tuning data and down rank the unsafe or low-quality ones.

This work has five reviewers. Four reviewers are positive to accept this work, while one reviewer is negative. Almost all reviewers think the problem is crucial, and the method is also motivated. Many experiments are conducted to verify the effectiveness of the proposed method.


In this regard, this work can be accepted.

**Additional Comments On Reviewer Discussion:**

Many reviewers raised concerns about the limited investigation of similar methods, more experiments, and the limited novelties of this work. After a discussion between reviewers and the authors during the rebuttal period, most of the concerns have been resolved and thus four reviewers are positive to accept it. One reviewer is still negative to accept it, and thus the authors are suggested to revise the paper based on these comments.

---

### Decision · Program_Chairs · 2025-01-22

Accept (Poster)